# Employment and Chronic Diseases: Suggested Actions for The Implementation of Inclusive Policies for The Participation of People with Chronic Diseases in the Labour Market

**DOI:** 10.3390/ijerph17030820

**Published:** 2020-01-28

**Authors:** Fabiola Silvaggi, Michela Eigenmann, Chiara Scaratti, Erika Guastafierro, Claudia Toppo, Jaana Lindstrom, Eeva Rantala, Iñaki Imaz-Iglesia, Andrew Barnfield, Alison Maassen, Matilde Leonardi

**Affiliations:** 1Neurologia, Salute Pubblica, Disabilità, Fondazione Irccs Istituto Neurologico Carlo Besta, 20133 Milan, Italy; michela.eigenmann@istituto-besta.it (M.E.); chiara.scaratti@istituto-besta.it (C.S.); erika.guastafierro@istituto-besta.it (E.G.); claudia.toppo@istituto-besta.it (C.T.); matilde.leonardi@istituto-besta.it (M.L.); 2Finnish Institute for Health and Welfare, 00271 Helsinki, Finland; jaana.lindstrom@thl.fi (J.L.); eeva.rantala@thl.fi (E.R.); 3Instituto de Salud Carlos III—“Carlos III” Institute for Health, 28029 Madrid, Spain; imaz@isciii.es; 4REDISSEC, Red de Investigación en Servicios de Salud en Enfermedades Crónicas, Instituto de Salud Carlos III, 28029 Madrid, Spain; 5EuroHealthNet, 1000 Brussels, Belgium; a.barnfield@eurohealthnet.eu (A.B.); a.maassen@eurohealthnet.eu (A.M.)

**Keywords:** chronic disease, employment, Toolbox, policy dialogue, inclusion, public health

## Abstract

In recent decades, the number of people living with one or more chronic diseases has increased dramatically, affecting all sectors of society, particularly the labour market. Such an increase of people with chronic diseases combined with the aging of working population affects income levels and job opportunities, careers, social inclusion and working conditions. Both legislation and company regulations should take into account the difficulties that workers experiencing chronic diseases may face in order to be able to formulate innovative and person-centred responses to effectively manage this workforce while simultaneously ensuring employee wellbeing and continued employer productivity. The European Joint Action “CHRODIS PLUS: Implementing good practices for Chronic Diseases” supports European Union Member States in the implementation of new and innovative policies and practices for health promotion, diseases prevention and for promoting participation of people with chronic diseases in labour market. Therefore, a *Toolbox for employment and chronic conditions* has been developed and its aim is to improve work access and participation of people with chronic diseases and to support employers in implementing health promotion and chronic disease prevention activities in the workplace. The Toolbox consists of two independent instruments: the Training tool for managers and the Toolkit for workplaces that have been tested in different medium and large companies and working sectors in several European countries.

## 1. Introduction

A growing share of the working population in Europe suffers from at least one chronic disease. Currently, the percentage of the chronically ill in the employment sector equals 19% of the labour force [1].

Parallel to this, in Europe, the rate of labour market participation of people over 55 years old is estimated to rise by 8.3% in 2020 and by 14.8% in 2060 [2].

Based on this evidence, the labour market participation of people with chronic diseases is becoming an essential issue to deal with, considering the reduction in labour supply, the shortage of skilled workforce and the pressure on the pension systems caused by the considerable ageing of the labour force [3]. Investing in healthcare and welfare policies for the working age population will increasingly become an imperative to guarantee the sustainability of social security systems [4].

An international study conducted by the Harvard School of Public Health (HSPH) [5] estimates that in the USA, in the next 19 years, there will be a $47-trillion loss of output due to chronic diseases and mental diseases and their impacts on healthcare and social security services, absenteeism and reduced productivity, persistent disability and reduction of income for the families involved.

Beyond the loss of output, further concerns relate to the impact of chronic diseases on the labour market dynamics [6], and the management of return to work of ill workers. 

Studies have shown that many countries apply a “quota system”, which is a percentage of jobs dedicated to people with chronic diseases, to help individuals to retain their jobs if they develop diseases while already employed, but this impacted negatively on the employment of job-seekers [7]. 

Furthermore, despite some innovations introduced by collective bargaining, work suspension is usually not adequate to manage long-term and severe diseases, as some chronic diseases can be, which would require instead flexible working hours and adapted work tasks for both workers and their caregivers [8].

Another aspect to consider in the work sector scenario is that the reduction of working activity (i.e., part-time) for workers with chronic diseases usually results in lower remuneration, at a time when they may incur in higher medical expenses.

Self-employed persons, such as artisans, small business owners and those who are economically dependent on a single principal/client will face even more insecurity if they become chronically ill.

These concerns can deeply impact on the career perspectives and professional growth potential of people that have or develop a chronic disease. It is then important to envisage innovative and individually-targeted welfare policies and measures able to promote new definitions of “productivity” and “workplace presence”, thereby helping to respond to and to integrate employees’ and employers’ needs. 

Faced with this evidence, the European PATHWAYS project (www.path-ways.eu), which identified the impact of chronic diseases on European socio-medical systems, has shown how the existing EU and national level legislations regarding inclusive work are not specifically addressed to people with chronic diseases [9,10,11]. Conversely, these legislations apply to specific population groups, such as people with disabilities, long-term unemployed and “fragile” groups, but do not necessarily answer the needs of people affected with one or more chronic diseases [12].

The 2006 UN Convention for the Rights of people with disability affirms that disability is the result of the interaction between a health condition and physical, “behavioural” and “environmental” barriers which prevent participation in society and thus inclusion in the labour market of people with disability in a condition of equality with others [13,14].

It is necessary to implement effective management policies, underpinned by supportive legislation, that include programs for inclusion, maintenance of and return to work for chronically ill workers. These policies can include professional education and retraining for workers who must change their role or duties. The implementation of initiatives promoting social inclusion cannot be dependent upon the good will or the indulgence of employers and human resources managers. It is a right, and as such, it needs to be operationalised in all contexts by a set legislative and political frame.

This is being done in many countries, such as Australia, Denmark, Finland, Netherlands, Sweden, United Kingdom and New Zealand, where policies promoting sick workers’ job retention and return to work have been successfully implemented [15].

The aim of this paper is to highlight how European Union Member States as well as relevant stakeholders can support the implementation of new or innovative policies and practices that promote the participation of people with chronic diseases in the labour market, specifically for their labour market inclusion, job maintenance and return to work following illness. 

For this reason, the paper focused on the results of the Joint Action CHRODIS PLUS in the area of employment and chronic conditions.

This Joint Action aims to promote the implementation of policies and practices for the management of chronic diseases by supporting the participating Member States and promoting the sustainability of actions. For this purpose, a total of 42 beneficiaries representing 20 European countries collaborate to implement pilot testing and generate practical lessons that contribute to the uptake and use of CHRODIS-PLUS results.

Within the Joint Action, a work package has been dedicated to Employment and Chronic Conditions. This work package is coordinated by the Fondazione IRCSS Istituto Neurologico Carlo Besta, FINCB, and by the Finnish Institute for Health and Welfare, THL, and its main objective is to improve work access, inclusion and participation of people with chronic diseases and to support employers in implementing health promotion and chronic disease prevention activities in the workplace.

## 2. Method

The methodology that was proposed for the elaboration of the two components of the Toolbox and that is presented in this paper used an iterative co-design process that integrated obtaining existing published scientific literature, qualitative research findings from relevant stakeholders and expert knowledge.

Scientific evidence was studied by conducting systematic literature studies on best practices for assessing and promoting employees’ wellbeing and health, preventing the development of chronic diseases, and fostering the work participation of employees with one or more chronic diseases [16,17,18,19]. Experiential data was collected through 45 interviews altogether, conducted with managers and employees from various industry sectors, as well as with occupational wellbeing and health professionals from six European countries.

The evidence identified was sequentially validated by the experiential knowledge of stakeholders collected through interviews. This process of work was generating concepts that iteratively resulted in outputs that were informing the design of the intervention (Table 1). The research team discussed and analysed the output(s) and critically reflected on the process. The diverse components of the Toolbox were assessed by key stakeholders resulting in a co-design process, through iterative testing and optimisation, that included both face-to-face meetings as well as tele-conferences.

## 3. Results

On the results emerged from scientific literature analysis and stakeholders’ interviews. an innovative tool, the *Toolbox for employment and chronic conditions* (Figure 1), has been developed and is an instrument that covers both the area of health promotion and chronic diseases prevention as well as the inclusion, integration, maintenance and return to work strategies for those who develop or have a chronic condition. The Toolbox in its globality is intended to represent a set of validated practices for the promotion of the participation of people with chronic diseases in the labour market across Europe, but also to provide inclusive environments for all workers. 

The Toolbox consists of two independent tools: the Training tool and the Toolkit. The Training tool was developed to help managers, employers, directors and human resources professionals to understand the benefits of inclusion, integration and return to work of people affected by chronic diseases. The Toolkit collects concrete, practical means for supporting wellbeing, health and work participation at the workplace. 

In detail, the Training tool *Promoting inclusiveness and work ability for people with chronic health conditions—a training tool for managers*, is based on a biopsychosocial approach to health. Thus, it is not disease-specific but based on and targeting human functioning, personal capabilities and chronic diseases commonalities as well as ensuring that the work environment is a facilitator and not a barrier to health and wellbeing [20]. 

The Training tool is composed of three sections and one appendix. In the first three sections, managers will learn how to measure and increase inclusiveness and workability of people with chronic diseases in public and private enterprises. The appendix includes information sheets on the most frequent chronic diseases and although it does not cover all chronic diseases, it provides a means for managers to understand how to cope with some of the most burdensome chronic conditions, i.e., breast cancer, multiple sclerosis, ischemic heart disease, depression, diabetes, back pain, migraine, stroke and chronic obstructive pulmonary disease. 

The Toolkit, *Fostering Employees’ Wellbeing, Health and Work Participation—Toolkit for Workplaces*, instead, collects concrete means through which workplaces can support the wellbeing and health, and enhance the work participation of all employees, regardless of their current work ability and health status. The Toolkit consists of seven domains (nutrition; physical activity; ergonomics; mental health and wellbeing; recovery from work; community spirit and atmosphere; smoking cessation and reduction of excess alcohol consumption), which are the areas that emerged from the interviews to be relevant for the overall wellbeing and health of workers. Every domain includes four different approaches that can be adopted to promote wellbeing, health, and work participation. The approaches target 1) knowledge and skills, 2) working environment, 3) organisation policies and 4) incentives. Under each approach, there are suggestions for concrete actions that a workplace can put into practice to improve employees’ wellbeing and health within the domain in question. Many of the proposed means are relatively effortless to put into action, meaning that their execution does not require major investments as regards personnel, time or material.

Both the Training tool and the Toolkit are undergoing an extensive European testing and their use is promoted in various countries across Europe (Italy, Finland, Spain, Germany, France, Hungary, Lithuania, the Netherlands) and working settings (medium and large enterprises). 

In order to understand the policy perspective on health promotion, disease prevention and innovative management of chronic diseases in the workplaces, a European Policy Dialogue on employment and chronic conditions was organised at the European Parliament in November 2019 to capture various perspectives of relevant stakeholders including policymakers, patients and employers/enterprises. 

The analysis of these perspectives allowed to have a broader vision of all the issues arising around this relevant topic and helped to collect stakeholders’ ideas regarding the implementation of the CHRODIS PLUS Toolbox for employment and chronic conditions.

The perspectives of chronic disease patients’ organisations’ representatives, employers and European policymakers provided an enriching area for further discussions and are summarised here below. 

### 3.1. The perspective of Chronic Patients’ Organisations

Employment is fundamental for staying connected to the community, maintaining skills and continuing to develop professionally. Many people with chronic illness continue working, or at least wish to do so, and working can have a beneficial impact on their recovery. 

With reasonable accommodation (e.g., part-time or modified work schedules; acquiring or modifying equipment; etc.) and adequate support at the workplace, those who wish to continue working can often continue to do so, resulting in a hugely improved quality of life that minimises the negative financial impact of chronic illness, the risks of social exclusion and poverty, and positively contributes to their mental health and overall wellbeing.

Most employees affected by chronic diseases claim to be more prone to discrimination and prejudices at work. A lack of support and understanding from colleagues and supervisors are important problems. Unfortunately, these problems of discrimination can result in workplace abuse, including bullying and harassment. A high number of absences from work and the lower productivity levels due to poor health are probably at the heart of these problems for some employees. It is important to establish appropriate employment-related rights and legislation for people with chronic conditions, to create supportive working environments and to cooperate across policy areas including health, education, employment and finance [21].

Measures to ensure that people with chronic conditions are better supported and encouraged to seek support when they need it, in educational and working settings, should be implemented.

### 3.2. The Perspective of Enterprises’ Representatives

In the last two decades, several European and International companies have begun to turn their attention to improving the health of their workers. Some employers have collected and distributed health insurance plans to their employees, with the hope that this information will result in better health outcomes. 

The focus of companies is on sustainable employment strategies, including postponing retirement, as a way to offset the rising cost of health care and demographic changes. Harnessing the potential of people with disabilities and chronic illness also helps businesses striving for a creative and innovative workforce [22]. 

These companies have a positive approach towards introducing retention or workplace adaption policies for employees affected by chronic diseases. Such companies tend to be those with a health or active ageing policy, or with a corporate culture characterised by trust and cooperative management styles. Typical measures applied by companies include adaptations to the quantity of work, work tasks, working equipment and providing training opportunities. 

It is fundamental that employers and enterprises are encouraged in such efforts to retain and support their employees’ health. Both national legislation and company regulations should allow for flexible management of employees living with chronic disease so that they can adapt continue working in a way that considers their disabilities.

It is also important that managers also are aware that chronic illnesses often present a range of symptoms and are often accompanied by other disorders and comorbidities, all elements which contribute to a compromised health status. 

A general lack of managerial knowledge on employees’ health and wellbeing influences leadership. This will only become more critical as the workforce ages and the burden of chronic disease rises. Investing on managers’ training in this topic is a way to promote the creation of an inclusive working environment.

Specific policies that seek to keep people with chronic diseases at work need to be implemented. These include, but are not limited to awareness-raising campaigns addressed at both companies and workers; specific guidance and training for work colleagues and managers; and assistance in workplace adaptation.

### 3.3. The Perspective of European Agencies

Across Europe, many workers affected by chronic diseases do not receive sufficient support, despite the fact that national laws often give them the right for reasonable adaptation of their workplaces. For instance, the European Foundation for the Improvement of Living and Working Conditions (Eurofound—https://www.eurofound.europa.eu/) shows that the prevalence of chronic diseases is increasing both in the elder and in the younger population. Only 30% of people with chronic diseases said they have had a workplace adaptation [7]. People with low education and low-paid occupation and without fixed long-term contracts are least likely to have their workplaces adapted. Employees with chronic conditions who have had workplace adaptations have much higher job quality than those without any adaptations. 

The Council Directive No. 2000/78/EC of 27 November 2000 asks Member States to formulate a coherent set of policies aimed at combating discrimination against groups, such as persons with disability. The Directive also provides that employers envisage “appropriate measures, i.e., effective and practical measures to adapt the workplace to the disability, for example patterns of working time, the distribution of tasks or the provision of training or integration resource” (art. 5 of Directive). 

Unfortunately, obligations contained in legislation are not as effective as they should be because they are not supplemented by adequate measures of work-life balance based on employability and the mutual adaptability of the parties concerned.

This allows workers with chronic diseases to continue participating in the labour market, and enterprises to promote better productivity and efficiency and employee loyalty, as well as to decrease costs (either direct or indirect).

## 4. Discussion

With this paper, we aimed to highlight how European Union Member States, as well as relevant stakeholders, can support the implementation of new or innovative policies and practices that promote the participation of people with chronic diseases in the labour market.

The results strengthen the need of a tool that can be a guide to better manage the employees with chronic diseases, to promote health and to prevent diseases. The Chrodis Plus Toolbox, in fact, could be very useful for employers, considering the responsibilities they have towards their workers that also include promoting and maintaining their employability. This clearly emerged from representatives of enterprises, of EU Institutions, as well as patients’ and civil society representatives during the European Policy Dialogue. 

Several points have been raised and suggest future activities. The first point is related to the opportunity to integrate workplace health in the ‘state of health in the EU cycle’. (i.e., Health at a glance report and country reports) The second point is that many aspects explored, in particular by the Training Tool for managers, can be taken up in the EU Semester process, proposing recommendations along these lines for reform to Members States in the areas of health and employment for example. The promotion of the TOOLBOX for employment and chronic conditions can be performed through activities with social partners. 

Finally, it is important that employers assess risks, particularly in relation to vulnerable workers, including those with CDs. It is not always known that there is this specific focus on vulnerable workers. Simple measures to keep vulnerable workers in place can often benefit the whole workforce. 

## 5. Conclusions

This paper focused on how EU Member States, employers, employees and others stakeholders can benefit from the results of the Joint Action CHRODIS PLUS in the area of employment and chronic conditions.

The Chrodis Plus Toolbox for employment and chronic conditions developed by the Joint Action CHRODIS PLUS can be used in the implementation of new policies and practices for participation of people with chronic diseases in the labour market. 

The international implementation and promotion of this instrument should be done through public awareness campaigns, events for national employers’ organisations and relevant stakeholders on issues related to employment and chronic diseases. Including the employment sector supports the approach of health in all sectors policies. This is only one step in the process of alerting Member States of the issues of employment and chronic disease and more work must be done by all relevant stakeholders.

## Figures and Tables

**Figure 1 ijerph-17-00820-f001:**
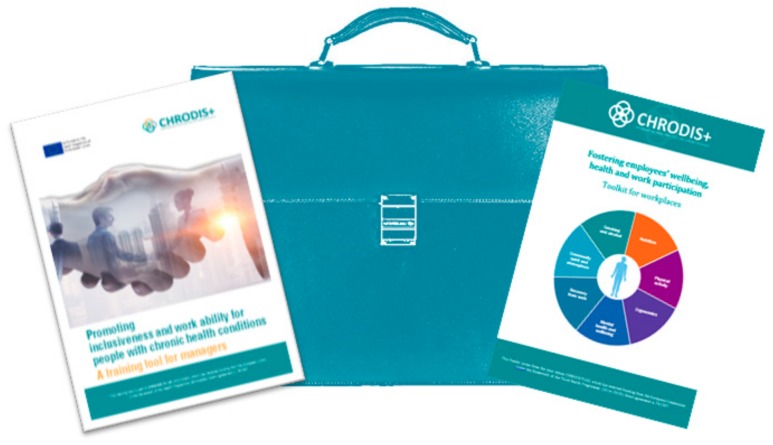
Cover of Toolbox for employment and chronic conditions.

**Table 1 ijerph-17-00820-t001:** Facilitators and barriers in the implementation of interventions facilitating work participation of employees with chronic health problems.

Facilitators	Barriers
- Employer’s motivation to foster work participation - Information available on best practices - Funding - Designating the responsibility for designing support practices to a certain quarter at the workplace - Finding flexible solutions for performing work tasks - Educating managers and teams - Culture of openness - Respect and trust in employees and their work ethic - Involving employees in planning work adaptations - Employees stating their needs clearly	- Lack of motivation among managers - Lack of knowledge among managers on ○ employees having health problems ○ the capability of individuals with chronic health problems to continue working ○ to what extent workplace can intervene in employees’ health concerns - Lack of funding and resources - Lack of communication within work community ○ Prejudice and stigma associated with health problems ○ Employees fear sharing their health problems Supporting RTW/work participation difficult if disease is not detected or treated early on

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
