# Peer review of "Employment and Chronic Diseases: Suggested Actions for The Implementation of Inclusive Policies for The Participation of People with Chronic Diseases in the Labour Market"

_ijerph, 2020, doi:10.3390/ijerph17030820_

Round 1
Reviewer 1 Report
This manuscript deals with an important topic, indicating the need of raising awareness about employment opportunities of people with chronic disease. It presents a short description of a Toolbox developed by authors, and it summarizes some conclusions from a European policy meeting. Although the content (including references) is of interest, this is not a scientific contribution to be published as an original Research document. Moreover, it seems that the Toolbox has not yet been evaluated, and no information e.g. about results of the 45 conducted interviews, is given. Simply proposing a Toolbox for Dissemination without further evidence is not an appropriate scientific strategy.
Apart from this crucial problem the manuscript has not been written with adequate care. For instance, p.4, line 164f. the sentence "Both this Toolbox and the toolkit are …" is incomplete, and the same holds true for Reference 21.
Author Response
To the Editor International Journal of Environmental Research and Public Health
Title of paper: Employment and Chronic Diseases: Suggested Actions for the implementation of inclusive policies for the participation of people with chronic diseases in the labour market
Corresponding author: Fabiola Silvaggi, PhD
Address of corresponding author:
Neurologia, Salute Pubblica e Disabilità
FONDAZIONE IRCCS ISTITUTO NEUROLOGICO CARLO BESTA
Via Celoria 11, 20133 Milano (Italy)
Telephone: +39.02.2394.3105
Email: fabiola.silvaggi@istituto-besta.it
Dear Editor and reviewers,
We were pleased to receive your response letter together with a set of useful comments on our manuscript entitled “Chronic Diseases & Employment: an overview of existing training tools for employers”, for which we really thank you.
We decided to undertake the revisions and tried to respond to each of the reviewers in the most appropriate way. Changes in the manuscript are in Track Changes" function, so that changes are easily visible to the editors and reviewers. Responses to reviewers are also reported in the present letter and in the specific form on the submission system.
Thank you for your attention, best regards
Fabiola Silvaggi on behalf of co-authors, Milano – 16/02/2019
Reviewer 1
Comments and Suggestions for Authors
This manuscript deals with an important topic, indicating the need of raising awareness about employment opportunities of people with chronic disease. It presents a short description of a Toolbox developed by authors, and it summarizes some conclusions from a European policy meeting. Although the content (including references) is of interest, this is not a scientific contribution to be published as an original Research document. Moreover, it seems that the Toolbox has not yet been evaluated, and no information e.g. about results of the 45 conducted interviews, is given. Simply proposing a Toolbox for Dissemination without further evidence is not an appropriate scientific strategy.
RESPONSE 1: We thank the reviewer for these important considerations. We totally agree and we reformulated the structure of manuscript and we added a section on the methodology.
Apart from this crucial problem the manuscript has not been written with adequate care. For instance, p.4, line 164f. the sentence “Both this Toolbox and the toolkit are …" is incomplete, and the same holds true for Reference 21.
RESPONSE 2: Thank you for this comment, we reformulated the sentence and we completed the reference 21
Reviewer 2 Report
In my opinion, this is an interesting and well-written manuscript on an important and actual issue as it is the employment of people with chronic diseases. I am happy to see that this issue has been addressed by Chrodis Plus Join Action and I hope that the final deliverable with the toolbox will be available soon for all the interested stakeholders. I have some few comments to improve the quality and to facilitate the reading of the manuscript.
- The manuscript should be revised in general for the correction of some grammar/punctuation mistakes.
- The manuscript does not follow the typical headings such as Introduction, Methods, Results and Discussion. It is not a problem for me, but I think it would be advisable to revise the manuscript to highlight somehow what was done in this study. For example, including a sentence with a summary of the actions performed in the joint action to address the problem, which could be interpreted as the methodology of the study (the development of the toolbox and the organization of the policy dialogue).
Abstract
- Lines 21-23. “more and more European workers find their job opportunities and income – and by extension – their social inclusion and working conditions”. I think this sentence is incomplete and does not make sense. Which idea did the authors want to transmit? Please, rephrase.
- Line 26. “to” is missing between “able” and “formulate”.
- Line 27. The second “to” left over from the sentence.
- Line 29. Replace the second “Chrodis” by “Chronic”.
- Line 32. I suggest explaining with a bit more detail in the Abstract the concept of “toolbox” developed. What does it consist of? Recommendations, reports…?
Section 1:
- I would suggest dividing this section into broader paragraphs (i.e. join some of the small sentences) to make the lecture more fluid.
- Lines 42-46. Revise and rewrite the two sentences not to repeat the ideas (“reduction in labour supply”, “shortage of skilled workforce”…).
- Line 56. What does the “quota system” consist of? Please, explain it in a few words to facilitate the reading.
Section 2:
- The authors should explain how the toolbox will be available for the interested stakeholders in the future (on demand, on EC website…).
- Lines 148-150. Delete this sentence as it repeats the same sentence of lines 142-144.
- Line 154: You talk about seven domains but it is difficult to see them separately inside the brackets; please use semi-colons to differentiate them from each other).
- Line 164-166. I think this sentence is incomplete. “The tools are undergoing an extensive… (what?) in various countries”.
Section 3
- Line 167. A comma or colon is needed between “Conditions” and “a challenge”.
- Line 174: comm? Delete
- Line 184: “e.g.” instead of “i.g.”
- Line 230: “For instance, the European…”.
Conclusions
- Line 248: This heading should be named as 4 instead of 5.
- The conclusions would benefit from a reorganization of the ideas presented. In my opinion, no references should be used in the conclusions.
- Line 253: “Joint Action” instead of “Joint Actions”.
Author Response
To the Editor
International Journal of Environmental
Research and Public Health
Title of paper: Employment and Chronic Diseases: Suggested Actions for the implementation of inclusive policies for the participation of people with chronic diseases in the labour market
Corresponding author: Fabiola Silvaggi, PhD
Address of corresponding author:
Neurologia, Salute Pubblica e Disabilità
FONDAZIONE IRCCS ISTITUTO NEUROLOGICO CARLO BESTA
Via Celoria 11, 20133 Milano (Italy)
Telephone: +39.02.2394.3105
Email: fabiola.silvaggi@istituto-besta.it
Dear Editor and reviewers,
We were pleased to receive your response letter together with a set of useful comments on our manuscript entitled “Chronic Diseases & Employment: an overview of existing training tools for employers”, for which we really thank you.
We decided to undertake the revisions and tried to respond to each of the reviewers in the most appropriate way. Changes in the manuscript are in Track Changes" function, so that changes are easily visible to the editors and reviewers. Responses to reviewers are also reported in the present letter and in the specific form on the submission system.
Thank you for your attention, best regards
Fabiola Silvaggi on behalf of co-authors, Milano – 16/02/2019
Reviewer 2
Comments and Suggestions for Authors
In my opinion, this is an interesting and well-written manuscript on an important and actual issue as it is the employment of people with chronic diseases. I am happy to see that this issue has been addressed by Chrodis Plus Join Action and I hope that the final deliverable with the toolbox will be available soon for all the interested stakeholders. I have some few comments to improve the quality and to facilitate the reading of the manuscript.
- The manuscript should be revised in general for the correction of some grammar/punctuation mistakes.
RESPONSE 1: We thank the reviewer for this consideration. We corrected grammar/punctuation mistakes of manuscript.
The manuscript does not follow the typical headings such as Introduction, Methods, Results and Discussion. It is not a problem for me, but I think it would be advisable to revise the manuscript to highlight somehow what was done in this study. For example, including a sentence with a summary of the actions performed in the joint action to address the problem, which could be interpreted as the methodology of the study (the development of the toolbox and the organization of the policy dialogue).
RESPONSE 2: Thank you for this comment, we totally agree and we reformulated the structure of manuscript and we added a section on the methodology.
Abstract
- Lines 21-23. “more and more European workers find their job opportunities and income – and by extension – their social inclusion and working conditions”. I think this sentence is incomplete and does not make sense. Which idea did the authors want to transmit? Please, rephrase.
RESPONSE 3: Thank you for this comment, we reformulated the sentence
- Line 26. “to” is missing between “able” and “formulate”.
RESPONSE 4: Thank you for this comment, we corrected it
- Line 27. The second “to” left over from the sentence.
RESPONSE 5: Thank you for this comment, we corrected it
- Line 29. Replace the second “Chrodis” by “Chronic”.
RESPONSE 6: Thank you for this comment, we corrected it
- Line 32. I suggest explaining with a bit more detail in the Abstract the concept of “toolbox” developed. What does it consist of? Recommendations, reports…?
RESPONSE 7: Thank you for this comment, we added in the abstract the informations on Toolbox
Section 1:
- I would suggest dividing this section into broader paragraphs (i.e. join some of the small sentences) to make the lecture more fluid.
RESPONSE 8: Thank you for this comment, we totally agree and we reformulated the paragraphs
- Lines 42-46. Revise and rewrite the two sentences not to repeat the ideas (“reduction in labour supply”, “shortage of skilled workforce”…).
RESPONSE 9: Thank you for this comment, we reformulated the sentences.
- Line 56. What does the “quota system” consist of? Please, explain it in a few words to facilitate the reading.
RESPONSE 10: Thank you for this comment, we have specified the sentences.
Section 2:
- The authors should explain how the toolbox will be available for the interested stakeholders in the future (on demand, on EC website…).
RESPONSE 11: Thank you for this comment, we have specified how the toolbox will be disseminated.
- Lines 148-150. Delete this sentence as it repeats the same sentence of lines 142-144.
RESPONSE 12: Thank you for this comment, we corrected it
- Line 154: You talk about seven domains but it is difficult to see them separately inside the brackets; please use semi-colons to differentiate them from each other).
RESPONSE 13: Thank you for this comment, we corrected it
- Line 164-166. I think this sentence is incomplete. “The tools are undergoing an extensive… (what?) in various countries”.
RESPONSE 14: Thank you for this comment, we reformulated the sentences.
Section 3
- Line 167. A comma or colon is needed between “Conditions” and “a challenge”.
RESPONSE 15: Thank you for this comment, we corrected it
- Line 174: comm? Delete
RESPONSE 15: Thank you for this comment, we corrected it
- Line 184: “e.g.” instead of “i.g.”
RESPONSE 16: Thank you for this comment, we corrected it
- Line 230: “For instance, the European…”.
RESPONSE 17: Thank you for this comment, we corrected it
Conclusions
- Line 248: This heading should be named as 4 instead of 5.
RESPONSE 18: Thank you for this comment, we edited it
Reviewer 3 Report
This is a very interesting paper on the CHRODIS Joint Action project which has potential to address the important issue of occupational adjustment for people with chronic diseases.
What it is not clear in the paper though is how the training tool and the toolbox will be used by the stakeholders, what kind of incentives there will be to promote use and how this use is going to be evaluated. Projects do finish at some point, Is there a plan for continuation in order to secure dissemination of the project's deliverables?
Also, I would suggest to run a thorough English language check as there are several points that need attention e.g. in the abstract it reads "The European Joint Action "CHRODIS PLUS: Implementing good practices for Chrodis Diseases". I suppose the authors want to say Chronic diseases.
Other parts that need attention are lines 64/65, 91/93, 162/163/ 171, 183 work not working, 206 adaptation not adaption, 228/229.
Author Response
To the Editor
International Journal of Environmental
Research and Public Health
Title of paper: Employment and Chronic Diseases: Suggested Actions for the implementation of inclusive policies for the participation of people with chronic diseases in the labour market
Corresponding author: Fabiola Silvaggi, PhD
Address of corresponding author:
Neurologia, Salute Pubblica e Disabilità
FONDAZIONE IRCCS ISTITUTO NEUROLOGICO CARLO BESTA
Via Celoria 11, 20133 Milano (Italy)
Telephone: +39.02.2394.3105
Email: fabiola.silvaggi@istituto-besta.it
Dear Editor and reviewers,
We were pleased to receive your response letter together with a set of useful comments on our manuscript entitled “Chronic Diseases & Employment: an overview of existing training tools for employers”, for which we really thank you.
We dedided to undertake the revisions and tried to respond to each of the reviewers in the most appropriate way. Changes in the manuscript are in Track Changes" function, so that changes are easily visible to the editors and reviewers. Responses to reviewers are also reported in the present letter and in the specific form on the submission system.
Thank you for your attention, best regards
Fabiola Silvaggi on behalf of co-authors, Milano – 16/02/2019
Reviewer 3
Comments and Suggestions for Authors
This is a very interesting paper on the CHRODIS Joint Action project which has potential to address the important issue of occupational adjustment for people with chronic diseases.
What it is not clear in the paper though is how the training tool and the toolbox will be used by the stakeholders, what kind of incentives there will be to promote use and how this use is going to be evaluated. Projects do finish at some point, Is there a plan for continuation in order to secure dissemination of the project's deliverables?
RESPONSE 1: We thank the reviewer for these important considerations. We totally agree and we reformulated the structure of manuscript adding information on dissemination.
Also, I would suggest to run a thorough English language check as there are several points that need attention e.g. in the abstract it reads "The European Joint Action "CHRODIS PLUS: Implementing good practices for Chrodis Diseases". I suppose the authors want to say Chronic diseases.
RESPONSE 2: Thank you for this comment, we corrected it
Other parts that need attention are lines 64/65, 91/93, 162/163/ 171, 183 work not working, 206 adaptation not adaption, 228/229.
RESPONSE 3: Thank you for this comment, we corrected them
Reviewer 4 Report
The manuscript is generally clear to the reader. It is, however, not very clear how it adds new knowledge to the field. Maybe the authors should consider clarifying what exactly they are trying to add and how this paper will be useful for policymakers and practice purpose.
Author Response
To the Editor
International Journal of Environmental
Research and Public Health
Title of paper: Employment and Chronic Diseases: Suggested Actions for the implementation of inclusive policies for the participation of people with chronic diseases in the labour market
Corresponding author: Fabiola Silvaggi, PhD
Address of corresponding author:
Neurologia, Salute Pubblica e Disabilità
FONDAZIONE IRCCS ISTITUTO NEUROLOGICO CARLO BESTA
Via Celoria 11, 20133 Milano (Italy)
Telephone: +39.02.2394.3105
Email: fabiola.silvaggi@istituto-besta.it
Dear Editor and reviewers,
We were pleased to receive your response letter together with a set of useful comments on our manuscript entitled “Chronic Diseases & Employment: an overview of existing training tools for employers”, for which we really thank you.
We decided to undertake the revisions and tried to respond to each of the reviewers in the most appropriate way. Changes in the manuscript are in Track Changes" function, so that changes are easily visible to the editors and reviewers. Responses to reviewers are also reported in the present letter and in the specific form on the submission system.
Thank you for your attention, best regards
Fabiola Silvaggi on behalf of co-authors, Milano – 16/02/2019
Reviewer 4
Comments and Suggestions for Authors
The manuscript is generally clear to the reader. It is, however, not very clear how it adds new knowledge to the field. Maybe the authors should consider clarifying what exactly they are trying to add and how this paper will be useful for policymakers and practice purpose.
RESPONSE 1: We thank the reviewer for these important considerations. We totally agree and we reformulated the structure of manuscript adding a part in discussion on how this paper will be useful for policy makers and practice purpose.
Round 2
Reviewer 1 Report
With the revisions made the manuscript is now acceptable for publication. It is clear that this is a document on work in Progress, and that scientific Evaluation of the Toolbox will be provided at a later stage. Based on this clarification I can revise my original judgment.
Author Response
To the Editor
International Journal of Environmental
Research and Public Health
Title of paper: Employment and Chronic Diseases: Suggested Actions for the implementation of inclusive policies for the participation of people with chronic diseases in the labour market
Corresponding author: Fabiola Silvaggi, PhD
Address of corresponding author:
Neurologia, Salute Pubblica e Disabilità
FONDAZIONE IRCCS ISTITUTO NEUROLOGICO CARLO BESTA
Via Celoria 11, 20133 Milano (Italy)
Telephone: +39.02.2394.3105
Email: fabiola.silvaggi@istituto-besta.it
Dear Reviewer,
We were pleased to receive your response letter together with positive comments on new version of our manuscript entitled “Chronic Diseases & Employment: an overview of existing training tools for employers”, for which we really thank you.
Thank you for your attention, best regards
Fabiola Silvaggi on behalf of co-authors, Milano – 22/01/2020